# Red Teaming the Rules: An Adversarial Approach to Legal Alignment

**Rui-Jie Yew & Greg Demirchyan**
Simons Institute for the Theory of Computing, UC Berkeley
{ryew, gdemirchyan}@berkeley.edu

## 1 Introduction

A core steering mechanism in a piece of regulation may lie not in what is written within it — the primal path it lays to follow its rules— but in what is left out of it — the dual path it lays to avoid them.

In 1696, King William III introduced a "window tax" on English residents: the number of windows a house had above ten would leave its residents susceptible to additional variable taxation. In response, residents covered up their windows and new homes were built with fewer windows (Oates & Schwab, 2015). In the 2010's, European seed makers increasingly used radiation to induce genetic mutations in their crops rather than classic genetic modification methods. By using radiation, seed makers were able to skirt European Union (EU) regulations whose scope was limited to the mutation of crops through methods like genetic engineering. The use of radiation over genetic engineering arguably induced even greater detrimental effects (Burk, 2016).

Today, the bypassing of legal rules continues across a wide range of domains–from tax law to product safety to intellectual property Katz (2011). Given a set of rules, regulated entities might opt not to break them or to comply with them, but to simply avoid them. At the same time, there are arguably benign cases in which the law may not apply. For example, it might make sense in certain instances for a company operating in California and incorporated in California to not be subject to certain aspects of Delaware corporate law. The particular form of legal avoidance we are interested in is that of *avoision*, which surrounds cases in which a certain kind of conduct which might comply with the letter of the law but subvert its spirit, or its goals — often through bypassing legal requirements entirely.

In this paper, we consider how red teaming methods for aligning AI systems can also be a generative frame for *legal alignment*, which we define as the alignment of legal texts to their policy goals. Our hope in this early-stage work is to demonstrate the relevance of AI alignment methods to regulatory design — in the sphere of legal drafting, and also in the realm of reasoning about and forecasting the behaviors of law-obeying AI agents (Tobey et al., 2024).

## 2 Background and Motivation

Red teaming is terminology introduced by the military and then re-purposed in the cybersecurity community to refer to a methodological way to surface potential vulnerabilities towards safer and more secure systems. In the AI community, it has been used to mean the surfacing of undesirable model behaviors. In this work, we are interested in a red teaming of the law against the undesirable behaviors it could permit. In so doing, we leverage the definition of *avoision* from legal scholars (Katz, 1996) as conduct that walks the line between "evad[ing] [a] law's intent or purpose" without it "actually constitut[ing] unlawful behavior" (Turner, 2001).

Scholars have described the identification of avoidance, encompassing avoision, as highly desirable in the legal sphere. As Julie Cohen writes, "Power interprets regulation as damage and routes around it" (Schneier, 2023). When it comes to tax law, Pistor (2019) notes that: "...shielding assets from taxes is one of the most sought-after coding strategies that asset holders covet. And lawyers ... are paid extraordinary fees to place them beyond the reach of creditors, including tax authorities". There has also been some work in modeling loophole behaviors computationally in (Qian et al., 2024).

Red teaming has also been relevant in for AI policy. Yew et al. (2025) "red team[s] AI policy" in a consideration of how the EU AI Act might be systematically subverted through a taxonomy of avoision. Tobey et al. (2024) discusses "legal red teaming" as "involv[ing] engaging legal professionals and technologists in a combined effort to prompt AI systems to produce outcomes that could pose legal and regulatory risks if provided to an end user." Ramakrishnan et al. (2025) discusses the problem of designing law-obeying agents, ensuring that these agents are faithful to legal goals. Hadfield-Menell & Hadfield (2019) calls on AI alignment researchers to leverage lessons in incomplete contracting, which lead to gaps in contracts and regulations.

Taking the analogy from Hadfield-Menell & Hadfield (2019) in the other direction, we frame AI alignment and red teaming techniques as relevant to the problem of legal alignment. This paper sets up a legal alignment problem using a similar red teaming component as in (Yew et al., 2025) based on the Self-RedTeam set-up in (Liu et al., 2025). Our formulation is an alignment problem because it goes beyond the task of red teaming (surfacing undesirable behaviors) and towards that of alignment: building more robust rules (analogously in the model alignment setting, this would mean building safer models). Unlike the red-teaming system presented in (Liu et al., 2025), which aims to align model behavior with model developers' goals, we aim to align legal texts with their underlying goals.

## 3 MODEL PROTOTYPE

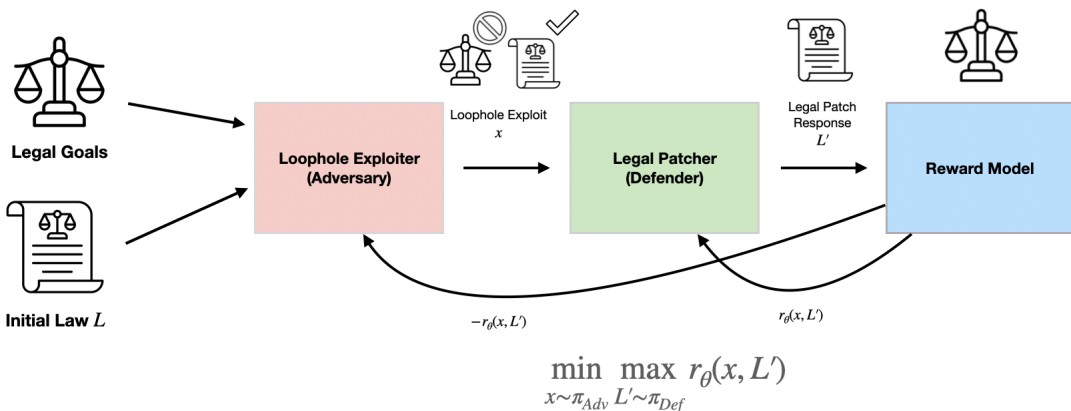

$$\min_{x \sim \pi_{Adv}} \max_{L' \sim \pi_{Def}} r_\theta(x, L')$$

We propose a prototype of a zero-sum legal alignment game with a conceptualization of each entity described in a colored box as an agent. The loophole exploiter aims to surface conduct that complies with the letter of the law but subverts legal goals. The legal patcher aims to "patch" the law such that the option for loophole exploitation is foreclosed. The reward model/ judge assesses how well the loophole exploit performs against the regulation based on its compliance with the law's letter, whether the exploit subverts the regulation's goals, and the degree to which the exploit subverts regulatory goals.

This process involves the legal goals as input, the initial legal text $L$ as input, a loophole exploiter which generates exploits, a legal patcher which generates a patched $L'$, and a reward model.

1. The loophole exploiter aims to surface conduct that complies with the letter of the law but subverts the laws goals, in line with the definition of *avoision* presented in Section 2.

2. The legal patcher aims to "patch" the law such that the option to exploit the loophole is foreclosed.

3. The reward model/judge assigns reward $-r_\theta(x, L')$ to the loophole exploiter and $r_\theta(x, L')$ to the legal patcher.

### 3.1 METHODOLOGICAL CONSIDERATIONS

**Legal Goals and Loophole Exploiter**   Legislative and policy goals are often specified regulatory design process. In EU regulations, the goals of a piece of legislation are commonly specified as part of an Act's recitals (Payne, 2024) while the Act's articles execute these goals. Certain laws in the United States also include a legislative findings/intent section which details the goals of a particular piece of legislation. "legislative intent" or "public policy" section. As part of the constitution, for example, the preamble details the overarching goals. There are certainly other sources that can be used in conjunction to determine legislative intent — for example legislative history, or other documents produced in the drafting of a particular piece of regulation. However, if we consider extending this model to a more systematic approach, it could make sense to input the equivalent of legislative intent/public policy goals or recitals.

**Reward Model and Legal Patcher**   The reward should be designed to capture how well the law at a given round $i$ of the game $L_i'$ captures the loophole exploit $\ell$, and, conversely, how well the loophole exploit slips under the radar of $L_i'$. This scoring scheme is non-trivial. Here, it is important to note how both specificity and ambiguity can give way to opportunities for loophole exploitation. As an example, the bright-line threshold for risk of $10^{25}$ FLOPs, like in the EU AI Act, can the splitting of computation across regulated entities through, for example, the use of federated learning (Hooker, 2024). On the other hand, overly "broad" (Sirman, 2023) rules, like Washington D.C.'s I-71 exception: "which allows a person 21 years of age or older to '[transfer to] another person 21 years of age or older, without remuneration, marijuana weighing one ounce or less'" (Sirman, 2023), created an industry where marijuana was added on as "gifts" to purchases, like \$550 pencils.

Being strategic about the degree of ambiguity and specificity can also be important for patching loopholes, but the reward function $L'$ might be encouraged to go towards "Don't be bad" to encompass every possible loophole behavior, or, at each round, $L'$ may simply patched with "Don't do $\ell$", where $\ell$ is the loophole exploit presented at a given round. Both approaches may be undesirable.

Indeed, a large part of legal design is finding the sweet spot between specificity and ambiguity. There is an analogous problem in the measurement of model alignment in red-teaming. The model can always refuse to answer a prompt it is given, in which case the reward model in (Liu et al., 2025) gives the response a score of $0$. We can take inspiration from Liu et al. (2025) and address the problem of finding a sweet spot between ambiguity and specificity based on the original legal text $L$ through, for instance, incorporating into the scoring system increased points for a faithfulness to the initial legal text (perhaps, to the initial legal text's length and other properties).

## 4 CONCLUSION

It may very well be impossible to draft rules without loopholes to exploit (Katz & Sandroni, 2017). However, perhaps unlike in the analogous security or AI system contexts under which a system that can defend against all attacker behaviors may be desirable, we may not necessarily want our regulations to be without loopholes. Loopholes can constitute some of the most important parts of a law. Sometimes, their exploitation can be a crucial driver of innovation (Burk, 2016; Sirman, 2023) and the subversion of power structures. Opportunities for later and further deliberation (McCray et al., 2010) can also be baked into regulations.

Rather than attempt to close all loopholes, through a system prototype like the one we propose, our hope is that the negative space of how a piece of regulation might be avoided is brought to the fore in how the path of its application is designed. Rather than steering behaviors towards legal goals through the primary path of where the rules apply, a legal alignment system presents a frame for steering behaviors towards legal goals against areas where rules may not apply. Further design of this red teaming model could include specific kinds of adversaries (borrowing from ideas in personateaming (Deng et al., 2025)) or use a scoring system that preferences the surfacing of loophole exploitations that cause more harm or that high-resource actors have incentives to pursue.

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
