# OpenReview forum: "Red Teaming the Rules: An Adversarial Approach to Legal Alignment"
_ICLR.cc/2026/Workshop/AFAA — AFAA 2026 Poster_

### Official Review · Reviewer_joeH · 2026-02-16
**Red Teaming the Law: An Adversarial Approach to Legal Alignment**

**Rating:** 3
**Confidence:** 4

**Summary:**

The paper considers how red-teaming methods for alignment of legal texts for policy goals can be useful.

**Strengths:**

The paper had an interesting background in terms of the setting up of the motivation. I like the idea of an alignment game, and wondered if the paper might want to consider other such games that may be cooperative, for example, and how might tradeoffs evaluate given different types of games. I was interested in why the model chose a zero-sum game versus another type of game with some relaxed flexibility parameter.
I found that the use of the policy and reward terminology was consistent with the field (e.g consistently used in areas such as reinforcement learning)

**Weaknesses:**

While I found that the introduction was very interesting, I think it could have been a bit more nicely blended into the rest of the paper. One suggestion would be to have less examples but spend more time on or two expository examples; for example, I really liked the “window tax” example, but there were too many examples after that, which muddied the analogy (at least for me, as a reader). I actually would have liked to see that first paragraph in the Background section as part of the introduction (perhaps even the last paragraph for the Introduction, as a way to segue into the Background and Motivation).
It would have been nice to separate what the explicit contributions are; this was a bit more difficult for me to extract from a first reading. There is a lot of exposition of previous works and direction, which the authors have done a great job of accumulating, but it was harder to extract what was novel about their approach.
I was also a bit skeptical of what it means to “patch” a loophole exploitation, and would (in a longer paper) like to learn a bit more about this method and its challenges. I really like the comment that “a large part of legal design is finding the sweet spot between specificity and ambiguity”, and found that starting with this concept for the paper might have been more engaging for me.
It would have been nice to have some initial evaluations.
I think generally although the paper contains some interesting insights and perspectives, it would really benefit from another iteration. In my mind, there are still some ideas that are not fully formed onto the page that could benefit from another rewriting of this article.

---

### Official Review · Reviewer_BPCh · 2026-02-22
**A timely and conceptually compelling framing of legal alignment as adversarial red-teaming, highly relevant to fairness in alignment procedures, but currently too preliminary without formalization or empirical validation.**

**Rating:** 3
**Confidence:** 3

**Summary:**

This paper proposes framing legal alignment as a red-teaming min–max game between a loophole exploiter and a legal patcher, with a reward model judging letter-compliance and goal-subversion. The contribution is primarily conceptual, outlining a high-level architecture and design desiderata for regulatory drafting tools. The framing is timely and directly relevant to fairness and alignment in agentic systems, but remains uninstantiated.

**Strengths:**

- Clear and compelling framing of legal alignment as adversarial red-teaming.

- Strong relevance to alignment procedures, governance, and fairness in agentic systems.

- Thoughtful discussion of the specificity–ambiguity trade-off in regulatory design.

**Weaknesses:**

- Lacks formal problem specification (action spaces, reward decomposition, constraints on patches).

- No implementation, dataset, case study, or empirical validation.

---

### Official Review · Reviewer_kEXK · 2026-02-23
**Red Teaming Based Legal Alignment via Adversarial Design**

**Rating:** 4
**Confidence:** 3

**Summary:**

The short paper proposes an adversarial framework for legal alignment, framing laws as systems that can be “red teamed” to surface loop-holes where compliance with the letter of the law undermines its policy goals. Drawing on analogies from AI red teaming, the authors introduce a zero-sum game between a loop-hole exploiter and a legal patcher, with a reward model evaluating the degree to which legal goals are preserved. The work positions this framework as a tool for stress-testing regulations during drafting and for reasoning about the behavior of law-complied AI agents under imperfect rules.

**Strengths:**

The strengths of the paper are listed as below:

1. This paper introduces a novel and compelling analogy between AI alignment and legal drafting.
2. The study clearly motivates the relevance of red teaming for regulatory design, not just enforcement.
3. The adversarial formulation is simple, intuitive, and extensible.
4. The approach is well-situated within both legal scholarship and AI alignment literature.
5. The methodology is appropriate scope and ambition for a short, idea-driven paper.

**Weaknesses:**

The weaknesses of the paper are listed as below:

1. The proposal is largely conceptual, with no empirical instantiation or case study.
2- Key components (reward modeling, legal goal specification etc.) remain underspecified.
3. Practical challenges in deploying such a system for real legislation are not fully explored.

---

### Meta-Review · Area_Chair_pxAr · 2026-02-28

**Recommendation:** Tiny/Short Papers Track
**Confidence:** 3

**Metareview:**

This paper proposes an adversarial red-teaming framework for legal alignment, modeling regulatory design as a g min–max game between a loophole exploiter and a legal patcher. Overall, the reviewers agree that the idea is timely and interesting, but they all point out that important technical details are missing or not clearly explained.

Reviewers like the original idea of comparing AI alignment to legal drafting, and find it relevant to governance and agent systems, especially the discussion of the specificity–ambiguity tradeoff and the zero-sum setup. However, they all point out that key technical details are missing, including clear definitions of the problem, reward modeling, and any concrete example or empirical validation. They also mention that the writing could be clearer, with a tighter structure, fewer examples, and a more explicit statement of the main contributions and what “patching” actually means in practice.

The main idea is interesting and well-timed, especially as we think about AI systems that must follow imperfect laws. However, the paper feels more like a position piece than a full research contribution, since the technical parts are not fully developed. I lean positive on the direction, but the paper would be much stronger with a clear formal setup, a simple worked example, and a sharper explanation of its main contributions.

---

### Decision · Program_Chairs · 2026-03-02

Accept (Poster)